Citation: *Molecular Systems Biology* 9:701
www.molecularsystemsbiology.com

# Promoters maintain their relative activity levels under different growth conditions

Leeat Keren[1,2,3], Ora Zackay[1,2], Maya Lotan-Pompan[1,2], Uri Barenholz[3], Erez Dekel[2], Vered Sasson[2], Guy Aidelberg[2], Anat Bren[2], Danny Zeevi[1,2], Adina Weinberger[1,2], Uri Alon[2], Ron Milo[3] and Eran Segal[1,2,*]

[1] Department of Computer Science and Applied Mathematics, Weizmann Institute of Science, Rehovot, Israel, [2] Department of Molecular Cell Biology, Weizmann Institute of Science, Rehovot, Israel and [3] Department of Plant Sciences, Weizmann Institute of Science, Rehovot, Israel
* Corresponding author. Department of Computer Science and Applied Mathematics, Weizmann Institute of Science, 234 Herzl Street, Rehovot 76100, Israel.
Tel.: +972 89346488; Fax: +972 89346488; E-mail: eran.segal@weizmann.ac.il

Most genes change expression levels across conditions, but it is unclear which of these changes represents specific regulation and what determines their quantitative degree. Here, we accurately measured activities of $\sim 900$ *S. cerevisiae* and $\sim 1800$ *E. coli* promoters using fluorescent reporters. We show that in both organisms 60–90% of promoters change their expression between conditions by a constant global scaling factor that depends only on the conditions and not on the promoter's identity. Quantifying such global effects allows precise characterization of specific regulation— promoters deviating from the global scale line. These are organized into few functionally related groups that also adhere to scale lines and preserve their relative activities across conditions. Thus, only several scaling factors suffice to accurately describe genome-wide expression profiles across conditions. We present a parameter-free passive resource allocation model that quantitatively accounts for the global scaling factors. It suggests that many changes in expression across conditions result from global effects and not specific regulation, and provides means for quantitative interpretation of expression profiles.

*Molecular Systems Biology* **9**: 701; published online 29 October 2013; doi:10.1038/msb.2013.59
*Subject Categories:* metabolic and regulatory networks; chromatin & transcription
*Keywords:* gene expression; growth rate; modeling; promoter activity; transcription regulation

## Introduction

Quantitative characterization of gene expression is a fundamental yet complex challenge. One of the major challenges stems from the dynamic nature of gene expression, whereby every gene can change its expression value across conditions. Although genome-wide analyses of expression are among the most commonly used methods in modern biology (Edgar, 2002), most studies produce lists of upregulated and downregulated genes, with limited focus on the numerical change in values. Here, we identify quantitative relationships between expression profiles under different conditions and observe a unifying behavior that simplifies our quantitative understanding of gene regulation.

Traditionally, gene expression research has focused on isolated genes and has generally shown that the transcriptional response is highly and specifically regulated. For example, upon exposure to lactose, bacteria respond by transcribing lactose-assimilating genes (Jacob and Monod, 1961). More recently, microarray and sequencing technologies have challenged this paradigm by enabling a genome-wide view of expression, and establishing that the responses to different conditions involve changes in expression of thousands of genes (Pedersen *et al*, 1978; DeRisi, 1997; Spellman *et al*, 1998;

Gasch *et al*, 2000, 2001; O'Rourke and Herskowitz, 2002; Boer *et al*, 2003; Saldanha *et al*, 2004; Tu *et al*, 2005; Lai *et al*, 2005; Shalem *et al*, 2008; Chechik *et al*, 2008; Brauer *et al*, 2008; Yassour *et al*, 2009; Costenoble *et al*, 2011; Tirosh *et al*, 2011). Such massive expression changes between conditions raise several fundamental questions. Primarily, it is unclear why the expression of so many genes changes even between conditions whose phenotypic differences appear to be minor. As one example, it is unclear why growing yeast on either glucose or its epimer galactose leads to detectable expression changes in over half of the yeast genome (Gasch *et al*, 2000; Chechik *et al*, 2008) even though only a few enzymatic reactions separate the two substrates.

Initial attempts to bridge the gap between specific regulation and the wide spread changes observed in the data suggested that specific responses actually encompass more genes than initially appreciated (Spellman *et al*, 1998; Gasch *et al*, 2000, 2001; Tu *et al*, 2005). More recently, it was shown that many changes in expression are correlated with growth rate (Pedersen *et al*, 1978; Regenberg *et al*, 2006; Castrillo *et al*, 2007; Brauer *et al*, 2008; Fazio and Jewett, 2008; Zaslaver *et al*, 2009; Klumpp *et al*, 2009; Levy and Barkai, 2009), as proposed decades ago by the Copenhagen school (Maaloe, 1969;

Ingraham and Ole Maaløe, 1983; Neidhardt, 1999), suggesting that they may result from global factors affecting many genes. Although several new works have attempted to incorporate global factors into gene expression models by analyzing synthetically constructed constitutive promoters (Klumpp *et al*, 2009; Scott *et al*, 2010; Gerosa *et al*, 2013), to date there is still no methodology to tease apart and decouple global and specific regulation. Therefore, it remains unknown what fraction of the gene expression changes observed upon a change in the growth condition can be explained by changes in global cellular parameters and which genes are specifically regulated. The ability to differentiate the two is critical for understanding the gene expression regulation. Moreover, except for isolated works (Van de Peppel *et al*, 2003; Bakel and Holstege, 2008; Islam *et al*, 2011), the importance of global changes in expression remains under-appreciated and is typically overlooked when analyzing high-throughput expression data. Such practice may lead to major misinterpretation of expression changes, as recently shown in Lovén *et al* (2012).

A second question invoked by the large magnitude of expression changes across conditions regards the combinatorics and complexity of gene expression programs. Considering all genes in all possible conditions, there is practically an infinite range of possible expression patterns that cells can reach. For example, yeast has ∼6000 potential transcriptional degrees of freedom as each of its ∼6000 promoters can potentially attain a different rate of transcription under each growth condition. It is intriguing to consider the degree to which this complexity is realized in cells. Indeed, genome-wide surveys of expression suggest that not all expression patterns are possible, as functionally related genes tend to be co-regulated (Pedersen *et al*, 1978; DeRisi, 1997; Gasch *et al*, 2000; Gasch *et al*, 2001; O'Rourke and Herskowitz, 2002; Boer *et al*, 2003; Saldanha *et al*, 2004; Lai *et al*, 2005; Chechik *et al*, 2008; Shalem *et al*, 2008; Brauer *et al*, 2008; Yassour *et al*, 2009; Costenoble *et al*, 2011; Tirosh *et al*, 2011). However, since most of these studies typically focus on the directionality of co-regulation (up or down) and not on the numerical change in values, to date the quantitative aspects of expression changes across conditions and their compliance with both classic (Maaloe, 1969; Ingraham and Ole Maaløe, 1983) and recent (Klumpp *et al*, 2009; Zaslaver *et al*, 2009; Scott *et al*, 2010) models of expression remain largely unaddressed. We aim to extend the existing qualitative description of co-regulated gene modules (Ihmels *et al*, 2002; Gasch and Eisen, 2002; Segal *et al*, 2003) and identify quantitative relationships between expression profiles under different conditions, thus reducing the space of possible expression patterns.

Here, we accurately measured the activities of ∼900 *S. cerevisiae* and ∼1800 *E. coli* promoters in 10 and 9 environmental conditions, respectively, using libraries of fluorescent reporters (Zaslaver *et al*, 2006, 2009; Zeevi *et al*, 2011). Notably, we found that most promoters (60–90%, depending on the pair of conditions compared) change their expression between conditions by a constant scaling factor that depends only on the conditions and not on the promoters' identity. Thus, although there is a major change in values for nearly all promoters, the relative activity levels are preserved. Accounting for global effects allows precise quantification of more

limited specific regulation—promoters deviating from global scaling. These can be organized into a handful of functionally related groups, such that within each group, promoters also preserve their relative activity levels across conditions in which they are activated. Hence, we can accurately describe 97% of the variability of the apparently complex promoter activity profiles across conditions using only several scaling factors. Finally, we present a parameter-free model that encompasses growth rate and specific gene expression and accounts for ∼90% of the observed variability in the global scaling factors. Our results provide a mean to decouple global and specific changes in activity between conditions, and a first quantitative characterization of the global response. They suggest that most changes in expression across conditions result from global effects and propose that proportional scaling is a major determinant of genome-wide expression profiles.

## Results

### Obtaining accurate measurements of promoter activity across different growth conditions

To obtain accurate measurements of promoter activity in yeast, we employed an experimental system based on the genomic fusion of promoters to fluorescent reporters (Zaslaver *et al*, 2006; Zeevi *et al*, 2011). We selected 867 native yeast promoters of genes that represent a wide variety of cellular functions, processes, and compartments (Supplementary Table S1). Although these genes cover only ∼1/6 of the *S. cerevisiae* genome, they cover the various Gene Ontology (GO) categories (Ashburner *et al*, 2000), promoter types (divergent/unique) (Saccharomyces Genome Database, available at: http://www.yeastgenome.org/), promoter architectures (OPN/DPN) (Tirosh and Barkai, 2008), and transcription regulation strategies (TFIID/SAGA-dominated) (Huisinga and Pugh, 2004). In addition, their combined expression represents ∼60% of the protein mass expressed in rich media (Wang *et al*, 2012) and thus accounts for much of the cellular activity under standard growth conditions (Supplementary Table S1, Supplementary material 1.1). We genomically integrated each promoter upstream of a yellow fluorescent protein (YFP) and used a robotically automated plate fluorometer to track the amount of reporter expression over time, in living cells, and across various growth conditions. Altogether, 859/867 (99%) promoters were successfully constructed. Simultaneous measurements of optical density (OD), indicative of population mass (Bremer and Dennis, 1987), enabled us to extract the doubling time of the culture and calculate the YFP production rate per OD unit per second (Methods, Zeevi *et al*, 2011), hereafter referred to as the *promoter activity* (Figure 1). These strains represent the largest library of native promoter-reporter fusions in eukaryotes to date.

The use of fluorescent reporters for measuring expression is a well-established approach (Bronstein *et al*, 1994; Kalir *et al*, 2001; Zaslaver *et al*, 2004; Newman *et al*, 2006; Ligr *et al*, 2006; Murphy *et al*, 2007; Cox *et al*, 2007; Gertz *et al*, 2009; Zeevi *et al*, 2011; Raveh-Sadka *et al*, 2012; Sharon *et al*, 2012), with several pronounced advantages. Unlike most current high-throughput techniques, which require cell lysis, fluorescence

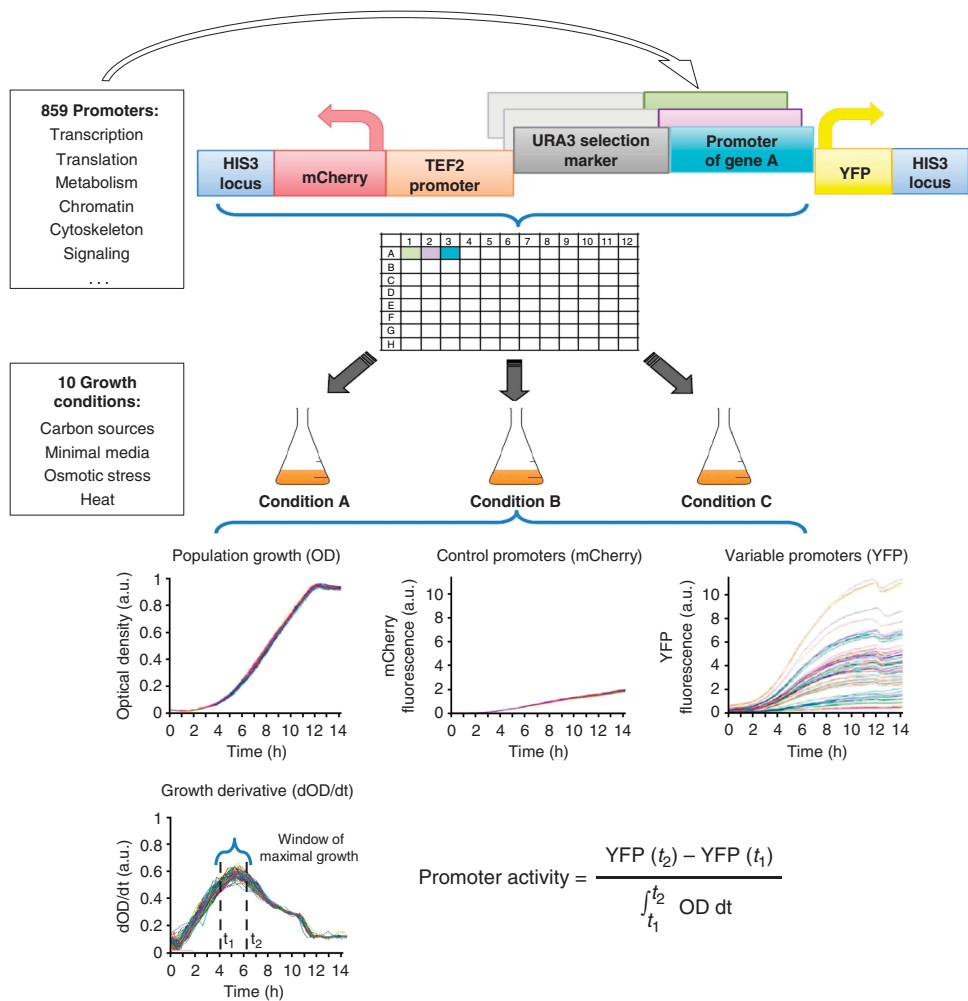

**Figure 1** Strain construction and promoter activity measurements. Illustration of our experimental system. The master strain into which we inserted the 859 different native yeast promoters that comprise our library is shown. Every promoter is integrated into the HIS3 locus upstream of a yellow fluorescent reporter (YFP). The master strain also includes a second mCherry reporter, driven by the same TEF2 promoter across all strains. Measurements are done in 96-well plates, where each well contains a different promoter strain, and cell density (OD), and YFP and mCherry fluorescence are measured along the entire growth curves of each tested growth condition. The low variation in OD and mCherry measurements across strains indicates that all strains grow similarly and that experimental variability is small, compared with the large span of the YFP values. For each strain in every growth condition, promoter activity was calculated as the YFP production rate per OD per second in the window of maximal growth (dashed black lines). Values from 1 to 6 replicate experiments were averaged to extract final promoter activities and standard deviations.

enables to perform live non-invasive imaging of the same cells over time with high temporal resolution. Accordingly, it does not require elaborate analysis and normalization techniques as required by other high-throughput measurement systems, such as microarrays or sequencing (Churchill, 2002; Marshall, 2004; Frantz, 2005; Bammler *et al*, 2005; Tang *et al*, 2007; Balázsi and Oltvai, 2007; Oshlack and Wakefield, 2009). This is especially important for our current study question as such normalizations may obliterate shared global effects (Lovén *et al*, 2012). It was also shown that fluorescent reporters provide highly precise and reproducible values (Bronstein *et al*, 1994; Kalir *et al*, 2001; Zaslaver *et al*, 2004; Ligr *et al*, 2006; Cox *et al*, 2007; Murphy *et al*, 2007; Gertz *et al*, 2009; Zeevi *et al*, 2011; Raveh-Sadka *et al*, 2012; Sharon *et al*, 2012). Here, we used replicate biological measurements to validate that our system provides highly sensitive and precise measurements for our set of promoters (CV ranging from 0.05 to 0.36 for promoters with high to very low activity, see

Materials and methods and Supplementary Figure S2); more reproducible than those obtained by microarrays, sequencing, or mass spectrometry (Supplementary Figure S3). Together, these features of the system are critical for the ability to detect and quantify the global and specific changes in expression reported here.

To make sure that our synthetic fluorescence measurements are representative of the true promoter activity (of the native gene in its native genomic location), we performed several analyses to gauge the integrity and accuracy of the system. We compared the promoter activities with quantitative real-time PCR measurements of 18 selected strains under two growth conditions, and confirmed that YFP levels are an accurate proxy for the corresponding mRNA levels ($R = 0.99$ and $R = 0.98$, Supplementary Figure S4A and B). In addition, we compared our promoter activity values with three microarray studies (Holstege *et al*, 1998; Shalem *et al*, 2008; Lipson *et al*, 2009), three RNA-seq studies (Nagalakshmi *et al*, 2008; Lipson

*et al*, 2009; Yassour *et al*, 2009), protein abundance obtained by immuno-tagged proteins (Ghaemmaghami *et al*, 2003), fluorescently tagged proteins (Stewart-Ornstein *et al*, 2012), mass spectrometry (De Godoy *et al*, 2008), and a curated data set of protein abundances integrated from five different data sets (Wang *et al*, 2012). In all of these comparisons, our promoter activity data correlated well with mRNA and protein abundance ($R = 0.72$–$0.81$ and $R = 0.57$–$0.74$ respectively, similar to the correlations between these data sets; Supplementary Figure S5), suggesting that promoter activity as measured by our system is a major determinant of these properties (Supplementary material 1.2). For further discussion of the experimental system, see Supplementary material 1.3.

## Most promoters' expression changes between conditions by a constant factor dependent only on condition and not on the promoter's identity

To compare promoter activities across conditions, we measured our library under 10 different environmental growth conditions, known to affect the expression of genes in the library (Materials and methods, Supplementary material 1.1, Supplementary Table S2). In line with observations using other methods, most promoters changed their activity levels between every pair of conditions, indicating that a major fraction of the genome responds to growth under different conditions (Supplementary Tables S3 and S4).

Next, we plotted promoter activities in every pair of conditions (Figure 2; Supplementary Figure S6). Strikingly, we found that in each such pair, most promoters change their expression between conditions by a constant factor that depends only on the conditions and not on the promoter's identity. This result indicates that although most promoters change their activity between conditions, they preserve their relative values. To quantify this global effect, we robustly fitted a scale line to the promoter activities of each pair of conditions, with the slope of the line ranging from 0.19 to 1 (arbitrarily setting glucose to 1, Materials and methods, Supplementary Table S4). We quantified the extent to which promoters adhere to the scale line by three independent methods: (A) Analysis of variance: We found that the scale line captures 80–99% of the variance in the data ($P < 10^{-45}$), depending on the pair of conditions compared. (B) We termed a promoter as behaving according to the global scale line if it deviated from it by less than three experimental standard deviations (Materials and methods). We found that between any two conditions, 60–90% of genes (depending on the pair of conditions compared) change expression by the same factor to within our relatively precise observation capacity. (C) Limiting the analysis to promoters that are moderately active in at least one of the two conditions being compared ($> 0.1$, for which our measurements yield lower STD values; Supplementary Figure S2), we found that 58–88% are within $\pm 30\%$ of the global scale line (Materials and methods). These independent analyses confirm that for most promoters, their activity in condition B is equal to their activity in condition A multiplied by a single number (the slope of the scale line between conditions A and B). We term this number the global scaling factor between conditions A and B and return to analyze it below.

In addition to the dominating global response, for each pair of conditions there remains a smaller subset of promoters that do not scale according to the global scaling factor (Figure 2, gray dots). We termed a promoter as condition specific if it is deviated from the scale line by more than three experimental standard deviations (Materials and methods), using glucose as a reference condition. We note that the relatively small sizes of the specifically responding groups does not appear to result from low representation of these promoters in the library, since our library was initially designed to represent different groups of genes (Supplementary Table S1; Supplementary material 1.1). Additionally, genes and environmental conditions were chosen together to include genes that are known to respond to the conditions (based on Gasch *et al*, 2000, 2001). Furthermore, we repeated the analysis, excluding known growth-related groups of genes, such as those involved in protein synthesis, and obtained very similar results (Supplementary Figure S10), indicating the robustness of our results with respect to input genes. We found that for each condition, its set of condition-specific promoters showed remarkable agreement with our understanding of yeast physiology and encompassed known co-regulated gene modules (Gasch and Eisen, 2002; Ihmels *et al*, 2002; Segal *et al*, 2003). For example, the specific response to galactose includes almost all the promoters whose corresponding genes belong to the galactose utilization pathway (8/9, $P < 10^{-3}$), while the specific response to amino-acid starvation is enriched for amino-acid metabolism (28/50, $P < 10^{-10}$, Figure 2C).

Our results agree with previous studies (Pedersen *et al*, 1978; DeRisi, 1997; Gasch *et al*, 2000, 2001; O'Rourke and Herskowitz, 2002; Boer *et al*, 2003; Saldanha *et al*, 2004; Lai *et al*, 2005; Chechik *et al*, 2008; Shalem *et al*, 2008; Brauer *et al*, 2008; Yassour *et al*, 2009; Costenoble *et al*, 2011; Tirosh *et al*, 2011), showing that a considerable fraction of the yeast genome changes activity levels between every pair of conditions. Here, the accuracy of our experimental system, together with the use of replicates, which allowed the construction of a reliable error model, enabled us, for the first time, to provide concrete means to tease apart global and specific regulation using the global scale line. This decoupling provides a novel reconciliation between the known specificity of gene expression regulation and the ubiquitous changes in expression observed in high-throughput data. It suggests that between conditions there exists a highly regulated, relatively small specific response augmented by many global changes. It further implies that in most cases, and in contrast to many common interpretations, changes in expression are not indicative of specific regulation.

In addition to decoupling global and specific changes in promoter activities between conditions, we provide a quantitative characterization of the global response. We find that global changes in activities between conditions are accurately captured by a single scaling factor, thus providing a first experimental validation for the theory of proportional scaling suggested decades ago by the Copenhagen school (Maaloe, 1969; Ingraham and Ole Maaløe, 1983). We note that the observed linearity is far from trivial, as different theoretical models exert different predictions regarding the expected behavior of unregulated promoters across conditions, including static, linear, non-linear, and promiscuous responses

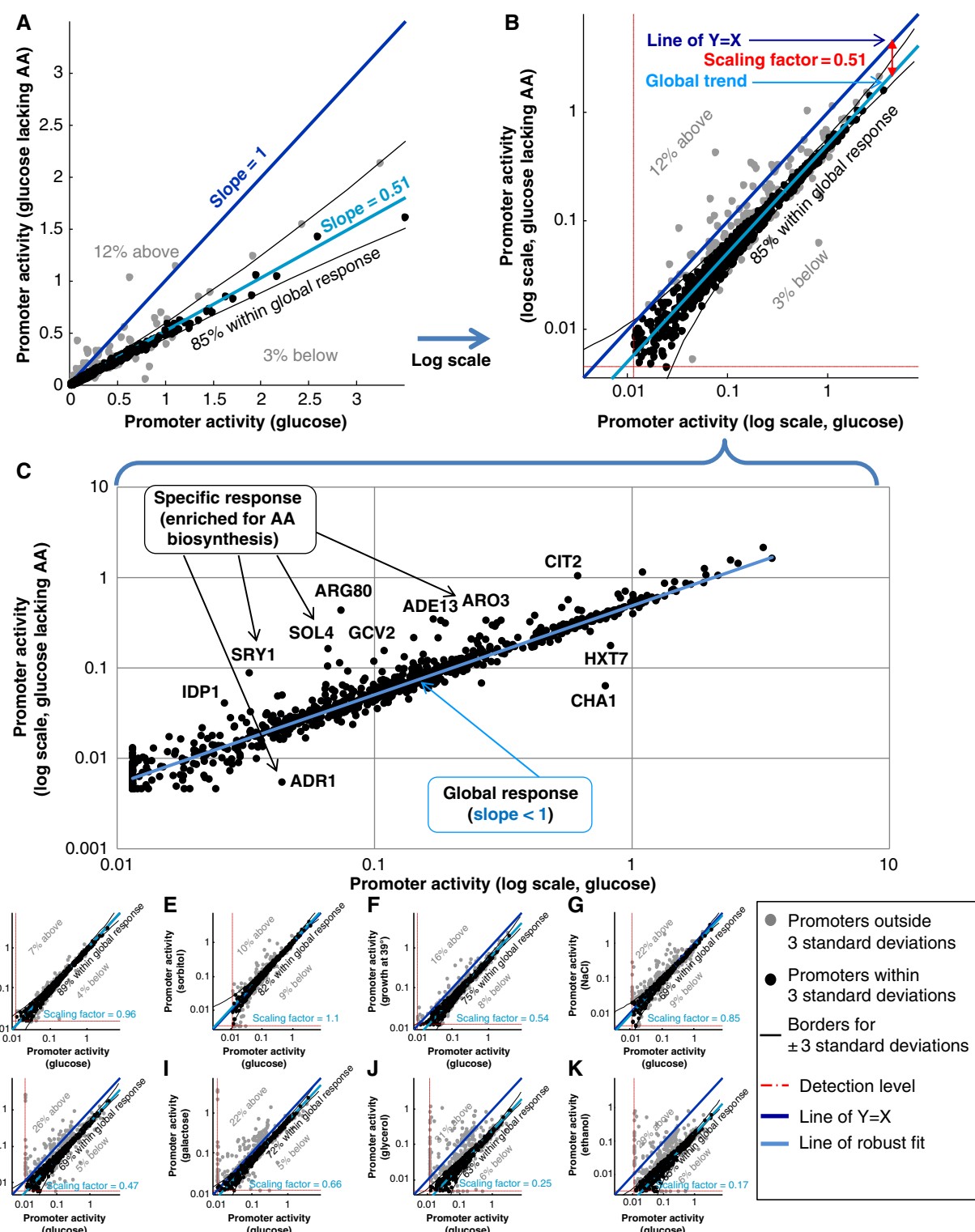

**Figure 2** Most promoters preserve their relative activity levels across conditions. (**A**) The promoter activity in glucose (*x* axis) and glucose lacking amino acids (y axis) is shown. Black lines represent three standard deviations of experimental noise around a robust linear fit to the data (cyan line) (Materials and methods). The slope of the robust linear fit represents the global scaling factor between the two conditions and is notably different than 1 (blue line), indicating that absolute values change between the conditions, but in a proportional manner. Promoters are colored black or gray depending on whether they fall within or outside the black lines, respectively. Dashed red lines indicate the lowest promoter activity level detected by our experimental system. (**B**) Same as in (**A**), but in logarithmic scale. Here, the scaling is reflected by the vertical shift between the blue and cyan lines. (**C**) Zoom-in on (**B**), showing the major global response and the identities of several specifically responding genes, all involved in amino-acid metabolism. (**D–K**) Same as in (**B**), but when comparing glucose (*x* axis) to fructose (**D**), sorbitol 1 M (**E**), growth at 39°C (**F**), NaCl 1 M (**G**), galactose lacking amino acids (**H**), galactose (**I**), glycerol (**J**), and ethanol (**K**).

(Supplementary material 1.4; Klumpp *et al*, 2009; Scott *et al*, 2010; Gerosa *et al*, 2013). Importantly, a prominent feature of the observed proportionality is that it preserves the ratio between most promoters across conditions. Specifically, it provides experimental evidence that complexes and pathways preserve the internal stoichiometry of their components, which may serve proper cell function. This finding reduces the degrees of freedom needed to characterize genetic programs, and opens the way for interrogation of the expression ratios being preserved.

## Condition-specific promoters follow alternative scale lines

We further wished to examine whether proportional scaling is also a prominent feature in the specific responses. Previous studies have shown that functionally related genes are usually co-regulated (Pedersen *et al*, 1978; DeRisi, 1997; Gasch *et al*, 2000, 2001; O'Rourke and Herskowitz, 2002; Ihmels *et al*, 2002; Gasch and Eisen, 2002; Boer *et al*, 2003; Saldanha *et al*, 2004; Lai *et al*, 2005; Chechik *et al*, 2008; Shalem *et al*, 2008; Brauer *et al*, 2008; Yassour *et al*, 2009; Costenoble *et al*, 2011; Tirosh *et al*, 2011). However, to date it was not determined whether individual members within these co-regulated groups maintain proportionality throughout conditions. To this end, we studied promoters which are off the scale line under two different conditions: when comparing condition A with B and also when comparing condition A with C. Comparing these promoters in conditions B and C shows that they too fall on a scale line (Figure 3A–C; Supplementary Figure S7). In most cases, its scaling factor is very close to the global scaling factor between conditions B and C (Figure 3C–E). This finding can be interpreted as follows: the off-line genes have altered specific regulation in condition A (e.g., due to specific transcription factors (TFs)), but in conditions B and C they are not differentially regulated and thus follow the global scaling along with most other genes.

In other cases, we found that functionally related condition-specific genes fall on a separate scale line (Figure 3G; Supplementary Figure S7). One interesting example is the comparison of growth on ethanol (a respiratory condition) and galactose (a semi-respiratory condition). Both conditions require activation of respiration-related promoters, yet to a different extent (Fendt and Sauer, 2010). We found that between these conditions, respiration-related promoters preserve their relative levels, but with a scaling factor that was nearly two-fold higher than the global scaling factor (0.5 versus 0.28; Figure 3F and G), consistent with the increased utilization of the genes associated with these promoters in ethanol. Thus, for most (87%) promoters, the relationship between their activities in ethanol and galactose can be characterized either by a global scaling factor that accurately accounts for 77% of the promoters, or by a respiration-related scaling factor that accurately accounts for another 10% of the promoters. The remaining promoters are specific to either ethanol or galactose.

Altogether, we found, as have many before us (Pedersen *et al*, 1978; DeRisi, 1997; Gasch *et al*, 2000, 2001; O'Rourke and Herskowitz, 2002; Gasch and Eisen, 2002; Ihmels *et al*, 2002; Segal *et al*, 2003; Boer *et al*, 2003; Saldanha *et al*, 2004; Lai *et al*, 2005; Chechik *et al*, 2008; Shalem *et al*, 2008; Brauer *et al*, 2008; Yassour *et al*, 2009; Costenoble *et al*, 2011; Tirosh *et al*, 2011), that functionally related genes are co-regulated. Our results additionally show that promoters within these functionally related groups tend to preserve proportionality, changing their expression according to a common scaling factor. Adding to previous observations on co-regulation, we show that the ratios between co-regulated genes remain constant. This finding provides another layer of structure to our quantitative understanding of the organization of the genome-wide response to different conditions, on top of the dominating global response. We propose that the prevalent specific response of yeast to different conditions is to alter the degree to which an entire group of co-regulated genes is activated, without changing the internal stoichiometry between the member genes of the group.

## Scaling factors accurately represent the entire promoter activity profile across conditions

Following the observations that proportional scaling underlies both global and specific responses in pairwise comparisons between conditions, we asked to what extent can the entire expression profile of yeast across conditions be represented by groups of genes that scale proportionally; and if so, how many such groups exist, and what are their sizes, scaling factors, and biological characteristics. To this end, we subjected the entire data set to k-means clustering with the cosine metric, as it has the property of preserving relative values within each cluster (Materials and methods). Notably, we found that partitioning our set of promoters into six well-separated clusters (Materials and methods, Supplementary Figure S8; Supplementary Table S3) accounts for 97% of the variance in our entire data set (Figure 4). This result indicates that if we could visualize our 10-dimentional expression space (in which every axis represents a different condition), promoter activities would not occupy the entire space, but rather adhere to only six lines (Figure 4; Supplementary Table S5, for more analysis see Supplementary material 1.5).

We performed several tests to assess the above clustering, both technically and biologically. First, we examined the clusters in terms of biological function by subjecting them to enrichment analysis of GO terms, TFs, promoter architectures (OPN/DPN), promoter types (divergent/unique), and transcription regulation strategies (TFIID/SAGA-dominated) (Materials and methods, Supplementary material 1.6). We found highly significant enrichments across all six clusters (Figure 4; Supplementary Tables S6 and S7). The first cluster contains most of the promoters (77%), and corresponds to promoters that scale according to the global scaling factor across all examined conditions. It includes all of the promoters in our data set whose corresponding genes represent components of the translation, transcription, chromatin, cell cycle, cytoskeleton machineries and most (85%) TFs. It is enriched for constitutive, TATA-less, TFIID-regulated promoters with an open chromatin architecture. This cluster represents promoters that are not differentially regulated across our set of conditions, and whose coordinated change in activity is mediated solely by global effects.

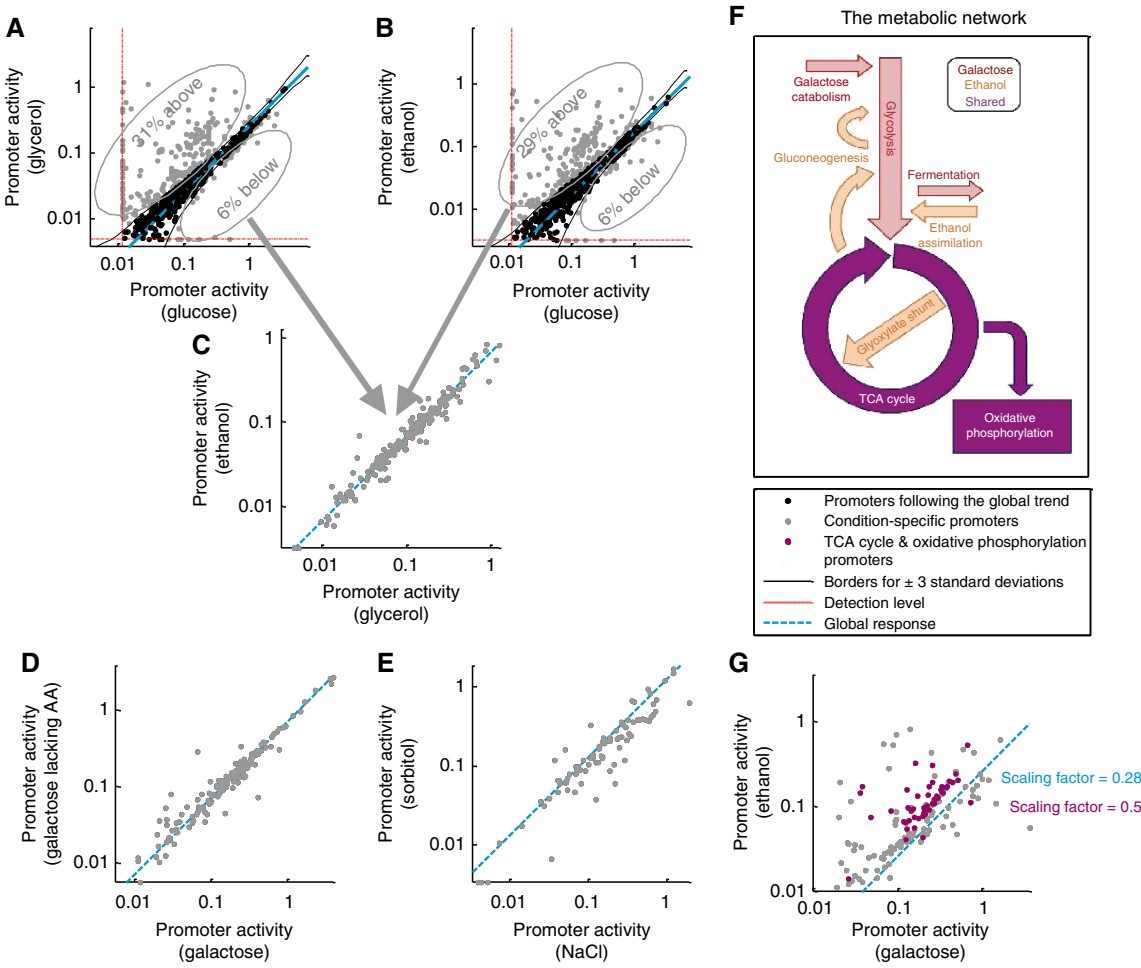

**Figure 3** Condition-specific promoters are functionally related and preserve proportionality across conditions in which they are activated. (**A**, **B**) Same as in Figure 2I and J, comparing promoter activities in glucose (*x* axis) and glycerol (*y* axis, **A**) or ethanol (**B**). (**C**) A comparison of shared condition-specific promoters (gray dots) from (A) and (B) between glycerol (*x* axis) and ethanol (*y* axis) is shown. Promoters display proportional activities, as indicated by their alignment to a straight line. The scaling factor for this subset of promoters is indistinguishable from the global scaling factor (cyan dashed line) with which the majority of promoters scale between these two conditions. (**D**) Same as in (**C**), for condition-specific promoters of galactose (*x* axis) and galactose lacking amino acids (*y* axis). (**E**) Same as in (**C**), for condition-specific promoters of NaCl 1 M (*x* axis) and sorbitol 1 M (*y* axis). (**F**) A schematic representation of central carbon metabolism in *S. cerevisiae*. Pathways are colored according to their literature-known activation in galactose (red), ethanol (orange), or both (purple). (**G**) Same as in (**C**), for condition-specific promoters of galactose (*x* axis) and ethanol (*y* axis). The subset of promoters belonging to the metabolic pathways activated in both conditions (purple in **F**) is colored in purple. These promoters display proportional activities as indicated by their alignment to a straight line. The scaling factor for this subset of promoters is different from the global scaling factor (cyan dashed line) with which most promoters scale between these two conditions.

The other clusters exhibit condition-specific responses in a subset of the tested conditions and are highly enriched for families of genes that are known to respond to these conditions and their known regulators (Figure 4; Supplementary Tables S6 and S7; Supplementary material 1.6). For example, cluster 2 is upregulated both in fully aerobic (ethanol and glycerol) and in partially aerobic (galactose and galactose lacking amino acids) conditions, and is highly enriched with respiratory genes ($15/16$, $P < 10^{-14}$). It also includes promoters of six TFs, and four of these (HAP4, CAT8, ADR1, and USV1) are major known regulators of respiration (Haurie *et al*, 2001; Segal *et al*, 2003; Tachibana *et al*, 2005; Fendt and Sauer, 2010). Thus, these clusters may represent complete regulatory units that are co-regulated across conditions in a manner that largely preserves their internal stoichiometry. Whereas the identification that these genes are co-regulated is not surprising, our finding that they preserve their relative activities is novel.

An important implication of our clustering results is that proportional scaling of promoter activities transcends the usual partition of promoters to housekeeping/condition-specific, growth-regulated/stress-regulated, open/closed NFR, TFIID/SAGA-dominated. Notably, even clusters 2–6, which displayed condition-specific behaviors in one or more conditions, were not differentially regulated between most conditions. Whether active or inactive, their scaling factors mostly coincided with the global scaling factor (Figures 3C–E and 4; Supplementary Table S5). This is consistent with the complementary observation that cluster 1 contains condition-specific promoters that are not differentially regulated across our tested set of conditions and therefore scale according to the global scaling factors across the entire data set (e.g., ER stress-associated proteins, which probably would have clustered separately if ER stress-inducing conditions were tested). These observations hint that the mechanisms

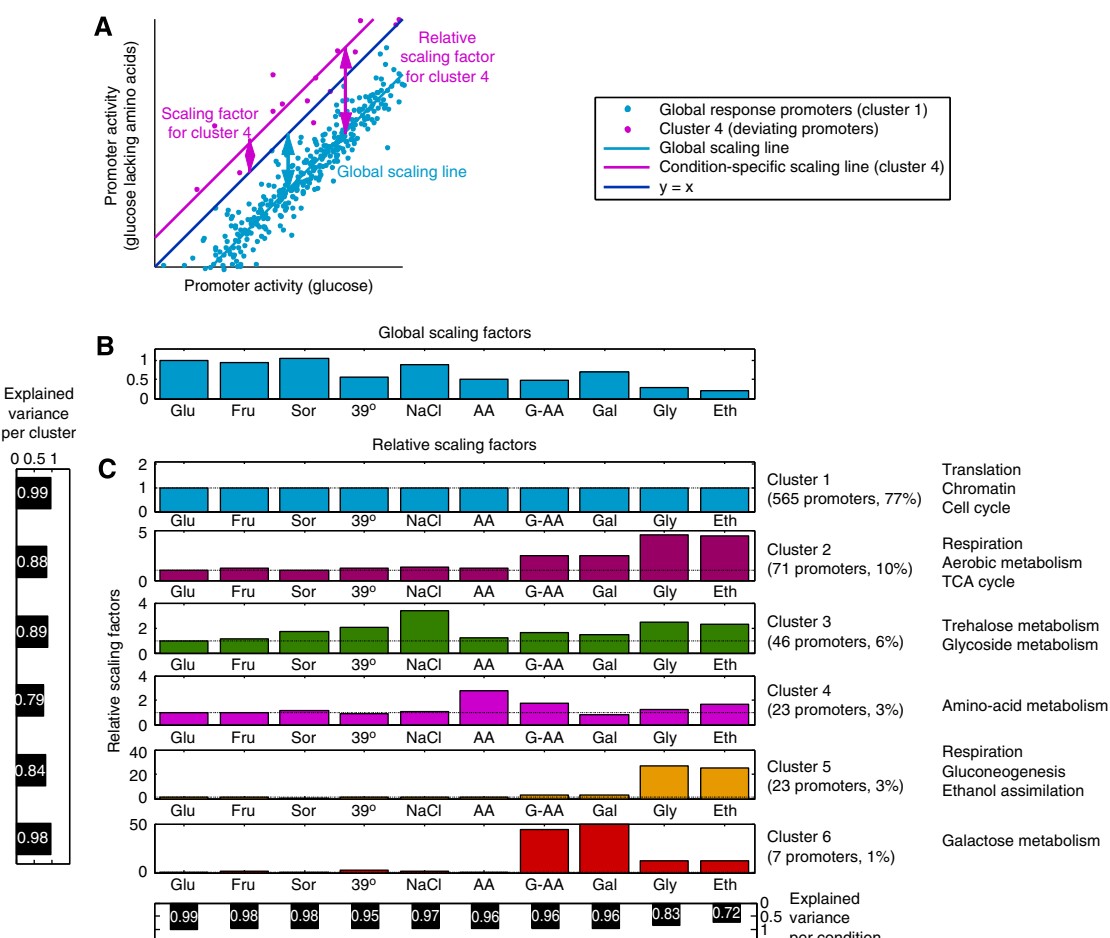

**Figure 4** Promoter activity profiles of every condition can be accurately described by a handful of scaling factors. (**A**) Promoter activities are compared between glucose lacking amino acids and glucose, as a zoom-in on Figure 2B. Promoters were clustered by their angle with the origin (Materials and methods). For each condition, both the global scaling factor, pertaining to the majority of promoters, and specific scaling factors were extracted. Since most genes scale with the global scaling, the relative specific scaling factors (relative to the global scaling factor) are of interest in the quantification of specific regulation. (**B**) Global scaling factors for all conditions, arbitrarily setting glucose to 1. Growth conditions abbreviations: Glu (glucose), Fru (fructose), Sor (sorbitol 1 M), 39°C (growth at 39°C), NaCl (NaCl 1 M), AA (glucose lacking amino acids), G-AA (galactose lacking amino acids), Gal (galactose), Gly (glycerol), Eth (ethanol). (**C**) Promoter activities across all conditions were clustered into six clusters in a 10-dimentional space (Materials and methods). In each condition, each cluster of genes scales by a unique scaling factor with low variability between the members of the cluster (left panel). Since all scaling factors (s.f.) in all conditions are relative to the global scaling factor of the condition, their values are indicative of relative activation (s.f. $> 1$), repression (s.f. $< 1$) or consistency (s.f. $= 1$) with the global scaling factor. The majority of promoters belong to the first cluster, scaling by the global scaling factor in all conditions. Selected GO terms that were enriched are indicated next to each cluster (for a more comprehensive list of enriched GO terms, see Supplementary Table S6). Using these scaling factors, we can accurately account for the variance in promoter activity in each condition (bottom panel).

responsible for global proportional scaling are not unique to a limited set of genes, promoter architecture, or transcription regulation strategy. Accordingly, known growth-related sequence motifs, such as binding sites for Rap1/Ifh1/Fhl1/ Sfp1/, RRPE and PAC (Hughes *et al*, 2000; Wade *et al*, 2001; Badis *et al*, 2008; Zhu *et al*, 2009) could not account for global scaling throughout all clusters. Thus, global proportional scaling of promoter activities is probably the result of a basic mechanism, shared across all promoter classes and architectures.

Next, we gauged the integrity and robustness of our clustering by means of cross-validation. For each of our conditions, we clustered the promoters based on all other conditions. We then used the activity values of only 10 predefined representative promoters in the tested condition to retrieve the scaling factors and thus predict the activity levels for the hundreds of other

promoters (Materials and methods). If our clustering truly captures the variation in the underlying data, then we would expect these predictions to match the measured activities. Indeed, using only 10 promoters, we obtained highly accurate predictions, whereby in every condition we can explain over 85% of the variance of the activities of at least 98% of the promoters (Figure 5; Supplementary Figure S9; Supplementary material 1.7). These results suggest that describing the response to a new condition may only require the description of the behavior of a relatively small number of clusters, representing the few degrees of freedom the cell actually utilizes.

Finally, we validated that our results were not determined by growth-related groups of genes, such as protein synthesis which are overrepresented in our library, by excluding these promoters from the data set and repeating the analysis (Materials and methods). We obtained very similar clustering

results with 95 % of the genes preserving their cluster identities (Supplementary Figure S10), indicating the robustness of our clustering with respect to input genes.

Our results present a first attempt to quantitatively analyze the transcriptional response of yeast across conditions in terms of proportional scaling. In addition to partitioning the genes into groups which are co-regulated, as is the usual practice in the field, here we show that individual members within these groups scale together and preserve proportionality. We find that the $\sim 900$ yeast promoters that we measured do not span the entire possible space of expression values, but rather adhere to a small number of scaling lines. This allows us to accurately describe the activity levels of the majority of promoters across conditions using only a handful of numbers representing the scaling factors of each of the different clusters. Moreover, we provide a first estimation for the relative magnitude of global and specific responses, showing that the former can account for much of the expression changes across many conditions. Together, our results suggest that gene expression profiles across different environmental conditions may exhibit less degrees of freedom and more structure than

previously appreciated. These findings could probably be largely extended to hold not only for promoter activities, but also for mRNA and protein levels based on the analysis of existing data sets (Supplementary Figures S5 and S11; Supplementary material 1.2).

## A simple passive resource allocation model accounts for a large fraction of the global scaling factors across conditions

Next, we asked what mechanisms can account for the quantitative values of the global scaling factors. Since enrichment analysis of promoter types, architectures, and regulation strategies did not provide mechanistic insights (Supplementary material 1.6), we examined two plausible theoretical models that result in proportional changes across promoters and compared their ability to explain the global scaling factor of each condition. In both models, we have no free parameters and we assume that promoter activity is an adequate proxy for protein production (Supplementary Figure S5; Supplementary material 1.2).

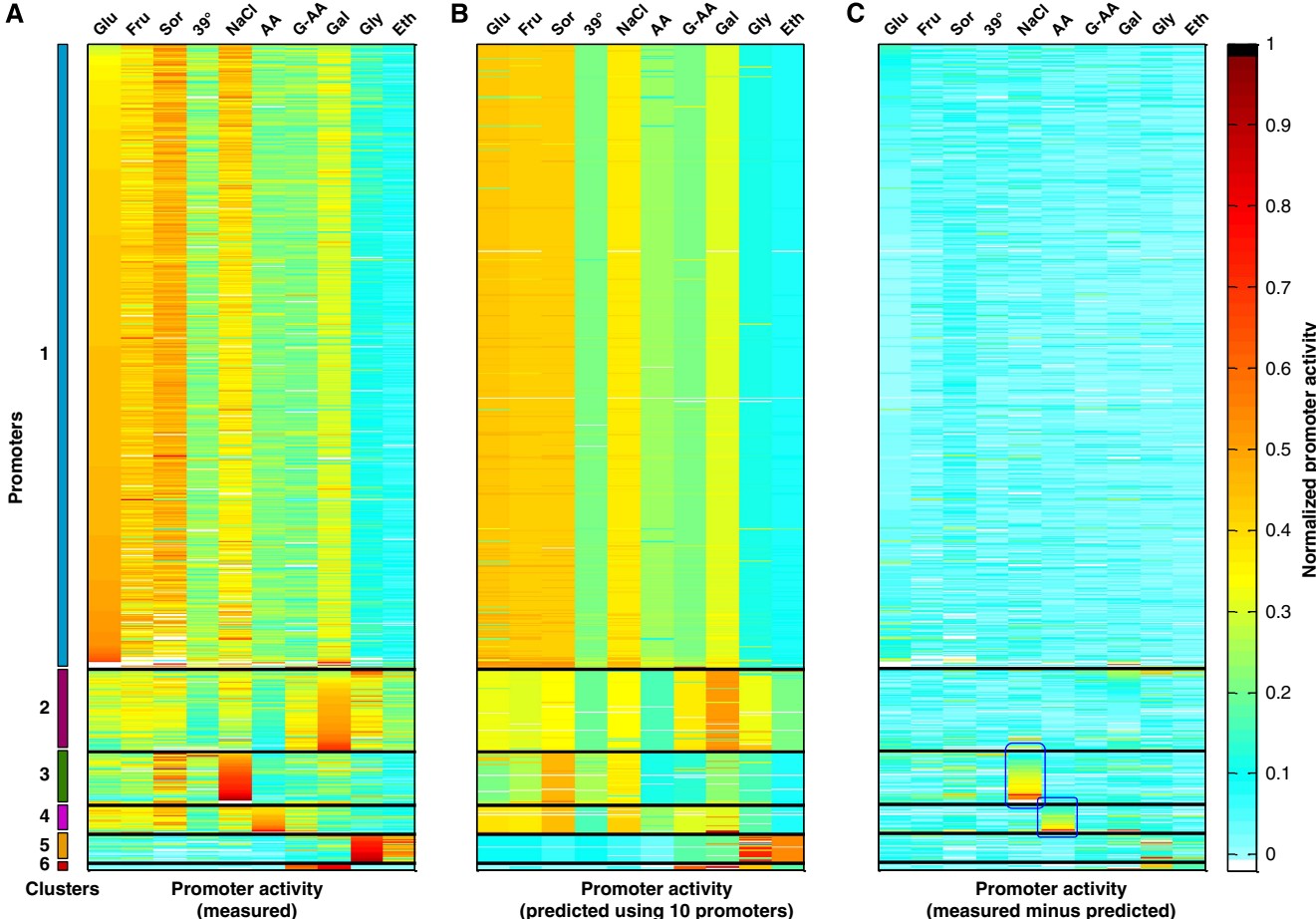

**Figure 5** Accurate prediction of promoter activities from several representative promoters. (**A**) Clustering of promoter activities where the value at row $i$ and column $j$ represents the activity level of promoter $i$ in the $j$th growth condition. Promoter activities were normalized across each row by dividing each entry by the vector norm such that values in each row sum to one. Growth conditions are abbreviated as in Figure 4B. (**B**) The same matrix from (**A**), but where promoter activities in each condition correspond to predictions. For each condition, predictions were generated using a clustering of the promoter activities of all other conditions and the values of 10 predefined representative promoters from the tested condition (Materials and methods). Predicted values were normalized as in (**A**). (**C**) The difference between the clustered matrices in (**A**) and (**B**) is shown. Blue boxes denote outlier clusters that are poorly predicted (e.g., the strong activation of cluster 3 in NaCl, Supplementary material 1.3).

Many genes were shown to change their expression in accordance with the growth rate (Pedersen *et al*, 1978; Bremer and Dennis, 1987; Neidhardt, 1999; Regenberg *et al*, 2006; Castrillo *et al*, 2007; Brauer *et al*, 2008; Fazio and Jewett, 2008; Zaslaver *et al*, 2009; Klumpp *et al*, 2009; Levy and Barkai, 2009; Scott *et al*, 2010), and we thus first tested whether growth rate provides a good approximation for the values of the observed global scaling factors. In this model, we expect that in conditions where cells divide twice as fast, the activity of most promoters would double, overall preserving their concentration. Formally, this model predicts that the global scaling factor from condition *A* to condition *B*, $S[A,B]$, will be given by:

$$S[A, B] = \frac{\tau[A]}{\tau[B]} \qquad (1)$$

where $\tau[A]$ and $\tau[B]$ represent the doubling time in conditions *A* and *B*, respectively. A possible mechanism for implementing such regulation is discussed in Supplementary material 1.4. This model entails that, per doubling time, the sum of activities of all unregulated promoters will be preserved across conditions (Figure 6A). We found that there is indeed a good correlation between the global scaling factors and growth rate (Supplementary Figure S12a) and that the scaling factors

predicted by this model deviate by $30 \pm 28\%$ from the measured scaling factors (3 of the 9 conditions are within 15%; Figure 6C).

Alternatively, in line with earlier models discussing differential allocation of resources between conditions (Ehrenberg and Kurland, 1984; Koch, 1988; Zaslaver *et al*, 2009; Molenaar *et al*, 2009; Scott *et al*, 2010), the overall promoter activity can be thought of as a fixed resource available to the cell per doubling time (Maaloe, 1969; Ingraham and Ole Maaløe, 1983). This is supported by our data, in which the total promoter activity per doubling time is relatively conserved (CV = 0.13, Supplementary Table S4, Supplementary Figure S12b). In each condition, this resource is differentially partitioned between the condition-specific genes, $G_{spe}$, and the globally responding genes, $G_{glo}$, where the exact partition is determined by the varying magnitudes of the specific response required in each condition (Figure 6B). Thus, the value of the global scaling factor will accommodate both changes in growth rate and the magnitude of the specific response in each condition. Formally, this model predicts that the global scaling factor from condition *A* to condition *B*, $S[A,B]$, is given by:

$$S[A, B] = \frac{\tau[A]}{\tau[B]} \frac{f[G_{glo}, B]}{f[G_{glo}, A]} \qquad (2)$$

where $f[G_{glo}, A] = \frac{P[G_{glo}, A]}{P[G_{glo}, A] + P[G_{spe}, A]} = \frac{\sum_{g \in G_{glo}} P[g, A]}{\sum_{g \in G_{glo}} P[g, A] + \sum_{g \in G_{spe}} P[g, A]}$

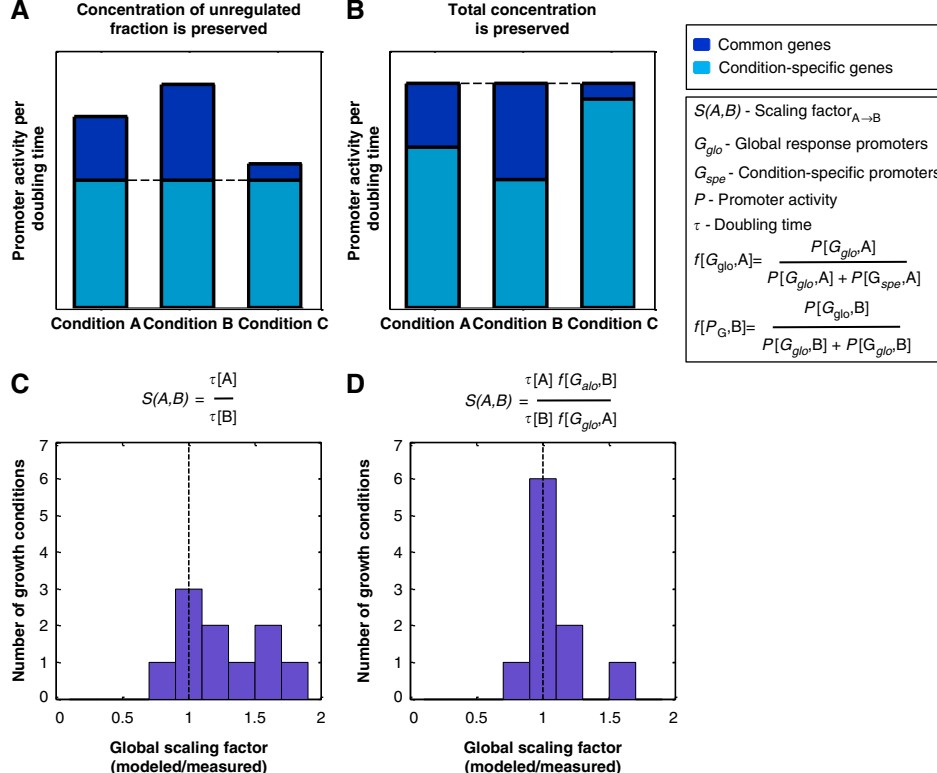

**Figure 6** A passive resource allocation model accounts for a large fraction of the global scaling factors across conditions. (**A**, **B**) Two proposed models for cellular invariants that result in the observed scaling factors: (**A**) Concentration of unregulated fraction is preserved. This model posits that growth rate accounts for the global scaling factors. It assumes that promoters that are not differentially regulated between conditions preserve their activity per doubling time. This entails the global scaling factor to be equal to the ratio between the doubling times of the compared conditions. (**B**) Total concentration is preserved. This model posits that both the growth rate and the magnitude of the specific response account for the global scaling factors. It assumes that the sum of activities over all promoters per doubling time is preserved. Promoters that are not differentially regulated between conditions will proportionally scale to accommodate the specific response in each condition. (**C**, **D**) Histogram of the ratio between the observed global scaling factors and those suggested by models (**A**) and (**B**) respectively.

represents the fractions of the total resources that are allocated to the global response promoters of conditions *A* and similarly for *B*. $G_{glo}$ represents the set of globally responding genes, $G_{spe}$ the set of condition-specific genes, and $P[g,A]$ and $P[g,B]$ represent the activity of promoter *g* in conditions *A* and *B*, respectively. Notably, this model achieves better explanatory power, deviating by only $12 \pm 19\%$ from the measured scaling factors (6 of the 9 conditions are within 15%; Figure 6D).

We thus propose this passive resource allocation model as a potential explanation for the global scaling of promoter activities between conditions. We suggest that an environmental change requires changes in the activation of condition-specific gene clusters, which in turn affects the amount of cellular resources available to all other genes. Most promoters in the cell are not specifically regulated under a given condition and therefore their activity level will passively vary based on the availability of global cellular resources, as captured by the global scaling factors, but still preserving their relative values (Supplementary Material 1.8).

We note that there remains a fraction of the proportional response unexplained by the resource allocation model. This may partly stem from inaccuracies in our estimates of the total activity of all promoters, since our library does not include all promoters. Additionally, this simplified model is per biomass unit as measured by OD and does not account for other global properties, such as cell size and macromolecular composition, which are known to vary in different conditions along with the doubling time (Schaechter, 1958; Bremer and Dennis, 1987; Neidhardt, 1999). Nevertheless, the accuracy with which this model matches the measured global scaling factors based on the doubling times and magnitude of the specific responses suggests that these factors are likely to be major determinants of the global scaling factors.

Together, global proportional scaling of promoter activities across conditions and the passive resource allocation model, which quantitatively accounts for the global scaling factors, can potentially shed a new light on the previously reported correlations between the expression of many genes and growth rate (Pedersen *et al*, 1978; Bremer and Dennis, 1987; Neidhardt, 1999; Regenberg *et al*, 2006; Castrillo *et al*, 2007; Brauer *et al*, 2008; Fazio and Jewett, 2008; Zaslaver *et al*, 2009; Klumpp *et al*, 2009; Levy and Barkai, 2009; Scott *et al*, 2010). First, in contrast to previous reports, which reported that different genes have different correlations with growth rate, here we suggest that different genes have the same quantitative coordination with growth rate (Supplementary Figure S13). Second, our results agree with classical theoretical models (Maaloe, 1969; Ingraham and Ole Maaløe, 1983), and show that the correlation with growth rate is not limited to a specific subset of genes, as was suggested by these works (Regenberg *et al*, 2006; Castrillo *et al*, 2007; Brauer *et al*, 2008; Fazio and Jewett, 2008). Rather, it is an inherent trait of expression resulting from global cellular constraints, and thus common to all genes in all conditions in which they are not differentially regulated. Finally, the resource allocation model suggests that global proportional changes in expression should extend far beyond the previously observed correlations with growth rate. We predict that such changes should occur not only in exponential growth under different environmental conditions, but in all cases where there are changes in global cellular resources. These include, for example, dynamic changes in the environmental conditions, mutations, different stages of the cell cycle, and different stages of the metabolic cycle. Indeed, a recent study showed that global regulation is growth rate dependent not only during steady state but also during dynamic changes in growth rate (Gerosa *et al*, 2013). Thus, scaling lines can potentially provide a conceptual framework for analyzing expression changes, whatever the condition.

## Promoter activities in *E. coli* scale proportionally between different growth conditions

To examine the generality of our results, we asked whether they also hold in other organisms. To this end, we measured promoter activities of $\sim 1800$ promoters of the model bacteria, *E. coli*, using a library of GFP-reporter strains (Zaslaver *et al*, 2006, 2009, Supplementary Figure S14A). Measurements were performed in nine growth conditions (Supplementary Table S8) and promoter activity was calculated by the rate of GFP production per OD unit, at the time of maximal growth, as described (Materials and methods, Supplementary Table S9; Supplementary Figure S14). In total, 969 promoters were active above background in at least one condition. Using replicate measurements, we validated that this system provides precise promoter activity values (Materials and methods, Supplementary Figure S13).

Notably, the results for *E. coli* were highly analogous to those obtained for *S. cerevisiae*, whereby between each pair of conditions 70–90% of the promoters preserve their relative activities (Figure 7B). As in *S. cerevisiae*, the global scaling factors for the different conditions span a wide range (0.51–1.68, arbitrarily setting M9 + glucose to 1, Supplementary Table S10). The scaling factors correspond to the growth rate and magnitude of the specific response predicted by the passive resource allocation model (85% of the variability explained; Supplementary Figure S16). Moreover, the specific responses are highly enriched with genes known to respond to the tested conditions (Supplementary Figure S17). Under conditions where a certain specific response was not differentially regulated, its promoters scaled according to the global scaling factor (Supplementary Figure S18). Here too, we can accurately predict the activity of most promoters in a new condition, using only a handful (five) of representative promoters (Materials and methods, Supplementary Figure S19).

Taken together, these results demonstrate that *S. cerevisiae* and *E. coli* exhibit similar global properties of transcriptional regulation, and that proportional scaling of promoter activity could be a basic and general trait across both prokaryotes and eukaryotes.

## Discussion

We quantitatively characterized promoter activities under different growth conditions. In agreement with previous studies (Pedersen *et al*, 1978; DeRisi, 1997; Gasch *et al*, 2000, 2001; O'Rourke and Herskowitz, 2002; Boer *et al*, 2003; Saldanha *et al*, 2004; Lai *et al*, 2005; Chechik *et al*, 2008; Shalem *et al*, 2008; Brauer *et al*, 2008; Yassour *et al*, 2009; Costenoble *et al*, 2011; Tirosh *et al*, 2011), we find that a

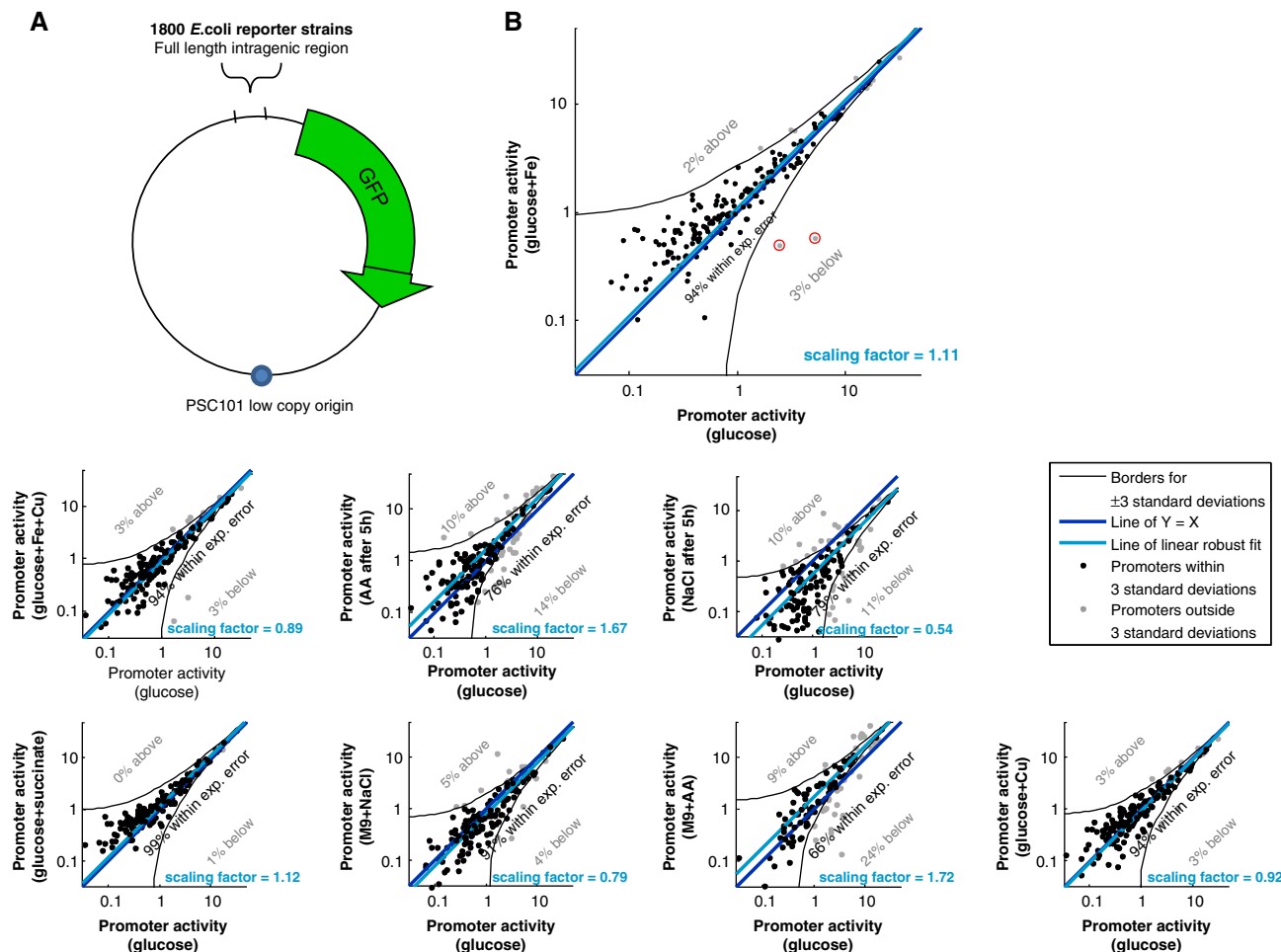

**Figure 7** Most *E. coli* promoters preserve their relative activity levels across conditions. (**A**) Reporter low-copy plasmid has a full-length intragenic region from *E coli* MG1655 driving the rapidly folding, non-toxic green fluorescent protein variant gfpmut2. Altogether, 1800 strains each corresponding to a different promoter were grown in 96-well plates in 37°C shaker incubator, and robotically moved every 8 min to a multi-well fluorometer for measuring GFP fluorescence and optical density over 24 h of growth. (**B**) Shown is a comparison of promoter activities glucose (*x* axis) and all other tested conditions (*y* axis) for the detectable *E. coli* promoters, as in Figure 2. Black lines represent three standard deviations around a robust linear fit to the data (cyan line). Promoters are colored black or gray depending on whether they fall within or outside the black lines, respectively. Scaling is reflected by the vertical shift between the blue and cyan lines.

considerable fraction of the yeast genome changes activity levels between conditions. We further show that most of this change is accurately captured by a global linear function whose single scaling factor corresponds to the change in growth rate and magnitude of the condition-specific response. When specific groups of genes are activated, these too change according to scaling factors, changing the degree to which the entire group is activated, while preserving the ratios between genes within the group. We show that the activity levels of all promoters in any condition can be accurately described using only a handful of numbers representing the scaling factors of each of the different groups, thus showing the quantitative manifestation of the hierarchical nature of gene expression. To the best of our knowledge, this is the first time that proportional scaling of expression was analyzed and demonstrated to have such a major role in genome-wide expression profiles.

Our results imply that in many cases, changes in expression are not necessarily indicative of specific regulation, as recently suggested by several studies (Klumpp *et al*, 2009; Scott *et al*,

2010; Lovén *et al*, 2012; Gerosa *et al*, 2013). The global proportional scaling that we detected provides concrete means by which to tease apart global and specific regulation. The ability to differentiate genes that are actively regulated from those that are merely responding to changes in global parameters is critical for the understanding of gene expression regulation, and may provide a powerful tool for focusing on 'functional biology'. Moreover, accounting for global effects allows precise quantification of specific regulation. Notably, we find that in each condition, only a relatively small and highly specific group of genes is actively regulated. The majority of expression changes across conditions, while real, seem to be the result of changes in global cellular factors and thus not informative about the regulation specific to the condition. This has broad practical implications, suggesting that global factors should be carefully taken into consideration when designing and analyzing studies that aim to understand gene expression regulation (Lovén *et al*, 2012).

Our work extends observations that the expression of many genes across conditions is correlated with growth rate

(Pedersen *et al*, 1978; Bremer and Dennis, 1987; Neidhardt, 1999; Regenberg *et al*, 2006; Castrillo *et al*, 2007; Brauer *et al*, 2008; Fazio and Jewett, 2008; Zaslaver *et al*, 2009; Klumpp *et al*, 2009; Levy and Barkai, 2009; Scott *et al*, 2010). Here, we considerably strengthen this correlation by further showing that different genes have the same quantitative coordination with growth rate (Supplementary Figure S13). Moreover, generalizing classical theoretical models (Maaloe, 1969; Ingraham and Ole Maaløe, 1983), we claim that the correlation with growth rate is not limited to a specific subset of 'growth-related' genes, as previously suggested (Regenberg *et al*, 2006; Castrillo *et al*, 2007; Brauer *et al*, 2008; Fazio and Jewett, 2008). Rather, it is an inherent trait of expression resulting from global cellular constraints, and thus common to all genes in all conditions in which they are not differentially regulated. The shared quantitative relation of many genes to growth rate greatly reduces the degrees of freedom of the expression program across conditions (as we show by our ability to accurately predict the activity of $\sim 70\%$ of the promoters in a new condition from measurement of only a single promoter), and simplifies our understanding of expression regulation. Proportional scaling paves the way for mechanistic understanding of the previously reported associations between growth rate and expression.

We also found that a simple parameter-free passive resource allocation model can account for the quantitative values of the global changes in promoter activities across conditions. If correct, then this model suggests that our findings regarding proportional changes in expression and their implications regarding specific and global modes of regulation extend far beyond the previously observed correlations with growth rate. We suggest that global proportional changes should occur not only in exponential growth at different growth rates, but in all cases where there are changes in global cellular resources. These include, for example, dynamic environmental changes, mutations, different stages of the cell cycle, different stages of the metabolic cycle, or different cell sizes. Thus, scaling lines can potentially provide a conceptual framework for analyzing expression changes, whatever the condition. It is interesting to speculate whether proportional scaling will be observed under such conditions and what will be the major determinant of the global scaling factors in these cases in which growth rate presumably has a minor role.

Our study opens many questions for future interrogation. Primarily, it will be interesting to learn how much of the proportionality observed for promoter activities is preserved by additional layers of regulation and perpetuates at the mRNA and protein levels, at both population level and in single cells. Additionally, it is interesting to consider what mechanisms are responsible for the observed proportionality and the coordination between promoter activity and growth rate. In bacteria, it has been suggested that cAMP (You *et al*, 2013), ppGpp (Magnusson *et al*, 2005), and use of alternative sigma factors (Klumpp and Hwa, 2008; Zaslaver *et al*, 2009) may contribute to the differential allocation of resources to different groups of genes. In yeast, transporters, cAMP, master growth regulators (e.g., *TOR*) and different ribosomal subunits have been suggested to perform a similar task (Broach, 2012). However, to date the molecular mechanisms underlying proportional responses, the identity of the limiting resources, and the role

that cell size, shape, and composition may have in this process remain unknown.

Finally, our observations regarding proportional scaling of condition-specific genes adds yet another layer of structure to our quantitative understanding of the organization of the genome-wide response to different conditions. We show that individual genes within co-regulated gene groups respond in a similar quantitative manner, thereby maintaining proportionality, that is, their relative expression values, across conditions. An important implication of this preserved proportionality is that complexes and pathways preserve the internal stoichiometry of their components across conditions. This greatly reduces the degrees of freedom needed to characterize a genetic program and quantitatively manifests the hierarchical nature of gene expression regulation. It invites further interrogation of the expression ratios being preserved and the numerical values of the different scaling factors.

## Materials and methods

A complete Materials and methods can be found in Supplementary material.

### Library design and construction

Promoters were chosen to cover a wide variety of cellular functions and processes (Supplementary material 1.1). Promoter sequences were defined as the genomic region between the translation start site (TrSS) and the end of the upstream neighboring gene. All strains were constructed as previously described (Zeevi *et al*, 2011), based on the genomic integration of promoter sequences into a common master strain, upstream of an YFP reporter (Supplementary material 2.1). Final strains were validated by sequencing, growth curves, and mCherry expression levels, and abnormal strains were removed. For a full list of promoters, primers, and sequences, see Supplementary Table S1.

### Promoter activity measurements

Cells were inoculated from frozen stocks into synthetic complete dextrose (SCD) (150 μl, 96-well plate) and grown at 30°C for 48 h, reaching complete saturation. Cells were then diluted 1:36 in fresh medium to a total volume of 180 μl and were grown at 30°C for at least 16 h in 96-well plates while being measured. Measurements were carried out every 20 min using a robotic system (Tecan Freedom EVO) with a plate reader (Tecan Infinite F500). Each measurement included optical density ($OD^{600}$), YFP fluorescence, and RFP fluorescence. Measurements of each plate at every growth condition were repeated 1–6 times. The growth media are outlined in Supplementary Table S2.

### Computing promoter activity levels

Basic analysis of measured OD, YFP, and RFP was done as previously described and included removal of strains with abnormal growth curves and RFP expression, subtraction of background levels of OD and auto-fluorescence, and smoothing of outlier measurements for each strain (Zeevi *et al*, 2011). Promoter activities were calculated as previously described (Zeevi *et al*, 2011; Raveh-Sadka *et al*, 2012) $pa = (YFP(t_2) - YFP(t_1))/ \int_{t_1}^{t_2} OD(t)dt$, with $t_1$ and $t_2$ defining a window of two doublings at the maximal growth rate (Supplementary material 2.4). The system's detection level was assessed by examining the distribution of promoter activity levels for a strain containing an RFP gene but no YFP gene. For each condition, $>30$ biological replicates of the strain were measured and fitted to a normal distribution (Supplementary Figure S1), and the 95th percentile of the distribution was taken to be the detection level. For each strain in every condition, we took the final promoter activity

levels to be the average of the strain across replicates. If this average was below the detection level, then we set the promoter activity to the detection level.

## Experimental variability

The relative error was estimated by the coefficient of variation (CV) of replicate measurements. Mean values across expression bins were used to estimate the CV of any promoter activity level, by linear interpolation (Supplementary material 2.6). The CV values ranged from 0.36 for very low promoter activity levels to 0.05 for high promoter activity levels (Supplementary Figure S2).

## Global scaling factors and error model

For all conditions, the global scaling factor represents the best robust fit to the data (Supplementary material 2.7). For each pairwise comparison, we determined the variance explained by the global scale line. The promoter activity values, $v(p)$, of each promoter $p$ was projected to the global scale line. Denoting the difference between the vector and its projection by $d(p)$, the variance explained by the clustering was calculated as $1 - \text{variance}(d(p))/\text{variance}(v(p))$. To obtain a $P$-value for the explained variance, for each comparison between conditions X and Y, we randomized the promoter activities of condition Y and quantified the variance explained by the original global scaling line, as described above. This was repeated 1000 times for each pairwise comparison.

For each pairwise comparison, we then determined which promoters behave according to the global trend between these two conditions using two separate methods: (A) We analyzed all data points above detection and estimated their probability to behave according to the global trend. For each promoter, let $(x, y)$ and $(sx, sy)$ denote its activity level and standard deviation in conditions $x$ and $y$, respectively. Denoting the scaling factor between the respective conditions by $a$, then promoters were defined as part of the global trend if $|ax - y| < 3sy$ or $|x - y/a| < 3sx$. (B) We restricted each pairwise comparison to promoters with an activity of $>0.1$ in both conditions. For such values, we found the average CV to remain constant at 0.05 (Supplementary Figure S2). For each pairwise comparison, we defined a promoter as part of the global trend if its value deviated by no more than 30% from its expected value according to the global scaling factor.

These two methods complement each other as the first is relative yet it enables the analysis of the entire data set, taking into account our different level of confidence in low and high values. The second is restricted to only parts of the data, yet it enables to determine absolute values. Both yielded similar results with $\sim 60$–$90\%$ of promoters (depending on the pair of conditions compared) changing between conditions according to the global scaling factor.

## Quantitative PCR analysis

Eighteen representative strains belonging to cluster 1 (RPB10, TEF1, DPM1, SEC61, SHP1, CDC10, RPS3, GLY1, RPL3, RPL33A, RPL8B, RPS7A, RPS11B, RPL4B, RPL28) or cluster 6 (GAL1, GAL2, GAL7) were inoculated from frozen stocks into SCD (150 µl, 96-well plate) and grown at 30°C for 48 h, reaching complete saturation. Cells were then diluted 1:36 in fresh medium and pelleted at mid-exponential phase. RNA was extracted using the EPICENTER Yeast MasterPURE RNA extraction kit, and cDNA was created using random hexamers (sigma). Quantitative PCR was performed by RT–PCR (StepOnePlus, Applied Biosystems) using a ready-mix kit (KAPA, KK4605). For each strain, measurements were performed in two sets of triplicates, measuring both YFP and RFP mRNA. Reported values are of mean YFP/RFP from nine replicates derived from three independent experiments (Supplementary material 2.8).

## Functional annotation and enrichment analysis

Sets of genes were assigned process, function, and cellular components according to the annotations from the GO (Ashburner *et al*, 2000). The significant representation of GO terms in the set was evaluated by Gorilla GO Term Finder (Eden *et al*, 2009) with a $P$-value threshold of $10^{-3}$. For TF analysis, we examined the distribution of known TF promoters (Badis *et al*, 2008; Zhu *et al*, 2009) across the different clusters. For enrichment analysis, promoters were classified as previously described according to their properties as: OPN/DPN (Tirosh and Barkai, 2008), SAGA-dominated/TFIID-dominated (Huisinga and Pugh, 2004), divergent/unique (this study, based on the Saccharomyces Genome Database, available at: http://www.yeast-genome.org/). $P$-values were computed according to the HG distribution and corrected for multiple hypothesis testing using FDR correction (Benjamini and Hochberg, 1995).

## Clustering promoter activities

To partition the promoters into clusters that preserve proportionality, we used K-means clustering with the cosine metric (defined by $(x, y) = 1 - \cos(\angle xOy) = 1 - \frac{x \cdot y}{\|x\| \cdot \|y\|}$, where $x$ and $y$ are vectors of promoter activity levels in a given condition and $O$ is the origin). The clustering was repeated 100 times with different random starting points and the clustering that minimized the sum of distances from the centers was chosen. The number of clusters, $K$, was determined as the largest $K$ for which the distance between any two centers is at least 0.05 (Supplementary Figure S8), thereby ensuring a minimal separation between any two clusters (Supplementary material 2.10). For generation of Supplementary Figure S10, this analysis was repeated excluding all ribosomal promoters.

## Variance explained by clustering

For each promoter $p$, its vector of promoter activity levels across conditions, $v(p)$, was projected to the center of the corresponding cluster. Denoting the difference between the vector and its projection by $d(p)$, the variance explained by the clustering was calculated as $1 - \text{variance}(d(p))/\text{variance}(v(p))$.

## Predicting promoter activity levels

We used the following scheme to predict promoter activity levels under growth condition $Y$ from measurements of several other conditions $x_1, \ldots, x_m$. First, the number of clusters $k$ for all promoters under the measured $m$ conditions was determined using above criterion. Then, the promoters were clustered by the k-means algorithm using the cosine metric. Denote the centers of the clusters by $c_1, \ldots, c_k$. A small number of representative promoters were chosen as the training set, and their promoter activity levels under the new condition $Y$ were used for the prediction task. For each cluster $t$, an extended center $\hat{c}_t$ of size $m + 1$ was calculated from the representative promoters that belong to cluster $t$. The activity level of a promoter under the new condition $Y$ was predicted to be $(\hat{c}_t(m+1)/\sqrt{\sum_{j=1}^{m} \hat{c}_t(j)^2}) \cdot \sqrt{\sum_{j=1}^{m} x(j)^2}$, where $x$ is the vector of activity levels for that promoter. The representative promoters were chosen such that an equal number of promoters were chosen from each cluster, which are closest (by the cosine metric) to the centers $c_1, \ldots, c_k$ of the relevant clusters (Supplementary material 2.12).

## *E. coli*

### Growth conditions

All media for bacterial growth were based on a defined M9 minimal medium (Supplementary material 3.1). Specific growth conditions and the respective growth rates in each condition are listed in Supplementary Table S8.

### Robotic assay for genome-wide promoter activity data

The library of reporter strains, each bearing a low-copy plasmid with one of *E. coli* promoters controlling fast-folding GFP (Supplementary

Figure S14A; Cormack *et al*, 1996) was previously described (Zaslaver *et al*, 2006). This library includes 1820 reporter strains that represent ~75% of *E. coli* promoters. Reporter strains were inoculated from frozen stocks into 96-well plates containing M9 minimal medium supplemented with 11 mM glucose, 0.05% casamino acids and 50 µg/ml kanamycin and grown overnight in a shaker at 37°C. All steps from this point were carried out using a programmable robotic system (Freedom Evo, Tecan Inc.). Overnight cultures were diluted 1:33 into M9 medium followed by a second 1:15 dilution into flat bottom microwell plate (nunc) containing one of the growth media (Supplementary Table S8) in a final culture volume of 150 µl. Bacteria were grown in an incubator with shaking (6 Hz) at 37°C for about 20 h. Every 8 min the plate was transferred by the robotic arm into a multiwall fluorometer (Infinite F200, Tecan) that reads the OD (600 nm) and GFP (excitation 480 (20), emission 515(10)). After 5 h of incubation, NaCl or casamino was added to the appropriate plates by automated pipetting.

## Computing promoter activity levels, detection level, experimental variability and error model

Promoter activity was calculated by the rate of GFP production per OD unit, as described above for yeast (Supplementary material 2.4) for the 3-h window around mid-exponential growth (Supplementary Figure S14). For conditions in which a compound was added to the media, promoter activity was calculated for the window of time after its addition. Background fluorescence was measured using a promoterless control strain in each plate. Promoter activities lower than 3 STDs above the mean background promoter activity were set to zero. In total, 969 promoters were active above background in at least one condition. Experimental variability was assessed as described above, using three replicate measurements in M9 glucose (Supplementary Figure S15) and error model was calculated as for *S. cerevisiae*.

Identification of representative promoters for predictions was done iteratively. At each iteration, we calculated the best linear sum of the representative promoter, which predicted the experimental data, and added an additional representative promoter, which contribute the most to predict the experimental data.

## Supplementary information

## Acknowledgements

This work was supported by grants from the European Research Council (ERC) and the US National Institutes of Health (NIH) to ES. ES is the incumbent of the Soretta and Henry Shapiro career development chair. RM is the incumbent of the Anna and Maurice Boukstein career development chair. We thank Michal Levo for helpful discussion.

*Author contributions:* LK and ES conceived the project. LK designed the yeast promoter library, and together with MLP constructed the strains. DZ and AW developed protocols for robotic strain assembly and constructed the yeast master strain. LK performed expression measurements and wrote the pipeline to extract promoter activity data. OZ performed analyses of clustering, prediction, and model evaluation. LK performed enrichment analysis and qPCR. LK, OZ, UB, RM, and ES conceived and evaluated the models. VS, GA, and AB performed measurements of *E. coli*, and ED analyzed the data. LK wrote the manuscript. UA, RM, and ES supervised and guided the research.

## Conflict of interest

The authors declare that they have no conflict of interest.

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
