## [Review Process File · Molecular Systems Biology]

Promoters maintain their relative activity levels under different growth conditions

Leeat Keren, Ora Zackay, Maya Lotan-Pompan, Uri Barenholz, Erez Dekel, Vered Sasson, Guy Aidelberg, Anat Bren, Danny Zeevi, Adina Weinberger, Uri Alon, Ron Milo, Eran Segal

Corresponding author: Leeat Keren, Weizmann Institute of Science

Review timeline:

Submission date:	05 August 2013
Editorial Decision:	29 August 2013
Revision received:	17 September 2013
Accepted:	27 September 2013

Editor: Maria Polychronidou

Transaction Report:

1st Editorial Decision

29 August 2013

Thank you again for submitting your work to Molecular Systems Biology. We have now heard back from the two referees who accepted to evaluate the study. As you will see, the referees find the topic of your study of potential interest and are in general supportive. They raise however a series of concerns and make suggestions for modifications, which we would ask you to carefully address in a minor revision of the present work.

Reviewer #2 makes a series of interesting suggestions for further reaching experiments. While we would definitely encourage inclusion of these analyses if the data are readily available, it is NOT a requirement for publication in this case. We would nevertheless ask you to at least include some of these ideas in the discussion as a perspective for future potential avenues of research.

We also attach a PDF file transmitted by reviewer #2, which elaborates on point #4 raised by this reviewer.

We would kindly ask you to add to Supplementary information the two central datasets analyzed in this study (~900 *S.cerevisiae* and ~1800 *E.coli* promoters x 10 and 9 environmental conditions, promoter activity and OD) so that others can reproduce your analysis and build upon this work. Please include this as an Excel or csv text file and upload it as 'dataset'.

REFEREE REPORTS:

Reviewer #1:

Review: Keren et al.

The authors examine the way gene expression, on a genome-wide scale, varies between different environmental conditions. In both *E. coli* and yeast, they find that most promoters change their expression by a constant global scaling factor that depends only on the conditions but not on the promoter's identity. Quantifying these global effects then allows them to characterize specific-regulation promoters that deviate from the global scaling. These are organized into few functionally related groups, and also adhere to scaling relations that preserve their relative activities across conditions. Thus, a small number of scaling factors are sufficient for accurately describing genome-wide expression profiles across conditions.

I recently reviewed this manuscript for one of the top journals, including a round of rigorous reviews and consequent changes to the manuscript. My final conclusion at the end of that process was that this is an excellent paper and should be published. I stand by this conclusion even more strongly for MSB: The subject matter, choice of questions, experimental and theoretical approaches, and main findings, would all be of great interest to readers of this journal.

Reviewer #2 :

The goal of this manuscript is to examine how promoter activities in yeast and bacteria change when the cells are grown in different conditions. Libraries of several hundreds of promoters fused to fluorescent reporters are used to estimate promoter activities in 10 different environmental conditions for yeast and 9 growth conditions for *E. coli*. The estimates are based on simultaneous fluorescence and optical density (OD) measurements using a Tecan plate reader. The first finding is that the majority of promoters change expression by a constant factor between conditions. The factor depends on the pair of conditions, and not on the specific promoter. The second finding is that the "exceptional" promoters (that do not change activities by the same factor as the rest of the genome) still change activities proportionally when comparing two conditions (B and C) to the same reference condition A. These observations allow the genome-wide prediction of promoter activities in new conditions after measuring them only a handful of genes. Two models are proposed to explain these observations. In the first model, promoters scale up or down their activities to keep in pace with different dilution factors arising from growth rates, meaning that the majority of gene products will alter their concentrations due to growth rate changes. In the second model, there is a condition-dependent differential allocation of total resources among promoters.

While condition-dependent coregulation of large gene groups has been observed before, this manuscript offers a new and surprising perspective, observing a constant condition pair-dependent scaling factor for most of the genome, and a very interesting scaling property across conditions even for the remaining outlier genes. The outlier genes are specific to one of the conditions investigated, validating the observation. Overall, this is an interesting and worthy contribution to Molecular Systems Biology. Nevertheless, I think that addressing the following comments would further improve the manuscript.

(1) It would be interesting to apply this method to some fluorescently tagged protein strains (from the genome-wide fusion library, see Nature 425(6959):686-91) using this method. While expression levels from such strains were compared with the promoter activities, it would be interesting to see how "effective protein synthesis rates" calculated with this method compare to the "promoter strengths".

(2) These are cell population average measurements. On the other hand, the fluorescent reporters allow single cell-level investigation. It would be interesting to know how the promoter strength variability changes from condition to condition. This is beyond the scope of this paper, but it would

be interesting to check in the future. Variability may actually contribute to the current results, if certain conditions "select" for a set of promoters to have low or high activity, then they will become over-represented in that condition, causing a shift in the mean promoter activity; see PLoS Comput. Biol. 2012 8(4):e1002480 for an example for non-genetic variant selection for a synthetic gene circuit.

(3) It would be interesting to see in the Discussion some further thoughts on the molecular mechanisms implementing the dilution factor model and the differential allocation model.

(4) The formula for bacterial promoter strength estimation differs from the formula used for yeast. What is the cause of this difference? And could this formula be derived? This is a question reaching back into 2002 and 2003 when the first papers from the Alon group using this formula were published. An attempt for a derivation is attached, should be edited, and should be included in the Supplementary Information. Specifically, there seems to be an implicit assumption about the constancy of cell number which is hard to justify in exponential phase. Could this be clarified?

(5) Most of the manuscript is about yeast, and the description of the *E. coli* data looks almost like an after-thought, despite containing data for a much larger fraction of the *E. coli* genome. Is there a reason for this?